# Practical method for RF pulse distortion compensation using multiple square pulses for low-field MRI

**Yonghyun Ha**[ID], **Kartiga Selvaganesan, Baosong Wu, Kasey Hancock, Charles Rogers, III, Sajad Hosseinnezhadian, Gigi Galiana, R. Todd Constable**[ID]*

Department of Radiology and Biomedical Imaging, Yale School of Medicine, New Haven, Connecticut, United States of America

* todd.constable@yale.edu

**Data Availability Statement:** All relevant data are within the manuscript.

**Funding:** The authors received no specific funding for this this work.

## Abstract

Since recovery time of the RF coil is long at low field MRI, the rising and the ring-down times of the square pulse are also long, which means the applied sinc pulse can easily be distorted from the changing amplitude. However, both the rising time and ring-down time can be calculated using $Q$-factor. Using this information, an RF square pulse were compensated by appending two square pulses before and after the RF pulse. The durations of these RF square pulses were calculated using the $Q$-factor. Since the amplitude of the sinc pulse changes continuously, a series of square pulses were applied to apply sinc pulse to the coil. The minimum number of square pulses and the amplitude of the square pulses were calculated. It was successfully demonstrated that the sinc pulse can be compensated using a series of square pulses. The more number of square pulses were used, the smoother sinc pulse was applied to the RF coil. The Q-factor was experimentally calculated from the ring-down time of a signal induced in a sniffer loop which was connected to an oscilloscope. The resulting Q-factor was then used to calculate both the duration and amplitude of the square pulses for compensation. Echo trains were also acquired in an inhomogeneous B0 field using the compensated RF pulses. In order to enhance the SNR of the echo trains, a pre-polarization pulse was added to the CPMG spin echo sequence. The SNRs of the echo signal acquired using compensated pulses were compared with those of signal obtained with uncompensated pulses and showed significant improvements of 61.1% and 51.5% for the square and sinc shaped pulses respectively.

## Introduction

The signal-to-noise ratio (SNR) in magnetic resonance imaging (MRI) is proportional to the static magnetic field, and is related to image contrast, resolution, and acquisition time. To achieve high SNR, higher magnetic field strengths generated from superconducting magnets (1.5–7 T) are commonly used for clinical and research studies. However, such magnets are expensive to purchase, site, install, and maintain (e.g. helium refills, cold head replacement),

**Competing interests:** The authors have declared that no competing interests exist.

such that access to MRI, especially in underdeveloped and developing countries is limited [1]. In response there has been a renewed push for alternative small footprint low field MRIs. Resistive and permanent magnets are candidates for low field MRI [2–7]. While moving to low field sacrifices some of the image quality achievable at high field MRI, there are still many benefits including: 1) the magnetic field generated by main magnet is safer as it reduces the risk of projectiles; 2) the specific absorption rate (SAR) associated with RF pulses is low (proportional to the square of magnetic field); 3) there can be greatly reduced acoustic noise from gradient coils since the forces on the gradients are proportional to the magnetic field strength; and 4) magnets can be made with a much smaller footprint and lower costs. The acoustic noise can be further reduced by using radiofrequency (RF) pulses for spatial encoding [8–11] which becomes much more feasible at low field whereas it is usually limited by SAR at high field MRI [12].

Accepting a lower main field also allows for the design of magnets with bigger bore sizes and/or open magnet designs, both of which can be helpful for people who suffer from claustrophobia in conventional high field MRI systems. An electromagnet system consisting of two elements with permanent magnets has been previously proposed [2]. Unlike typical MRI systems, this system generates inhomogeneous Bo magnetic field. Inhomogeneity of the $B_0$ field makes spin dephasing faster, leading to rapid loss of coherence at the polarizing field strength. However, to mitigate the inhomogeneous Bo field, the design utilizes an electromagnet the main field is ramp-able, allowing polarization to take place at high field (thereby maximizing signal), while signal acquisition is performed at low field [3–5]. In the presence of any gradient, including those that arise from non-uniform $B_0$ fields, selective spin excitation can be performed in a manner identical to that using slice select gradient coils by adjusting both resonance frequency and bandwidth of the RF pulse [2].

However, at low fields, the recovery time for RF coils is long because of the narrow bandwidth of the coil. It is proportional to the quality factor of the coil (*Q*-factor), and inversely proportional to the resonance frequency [13]. After switching off the RF transmit pulse, stored energy in the coil during transmit takes some time to de-energize, known as the coil ring-down time [14]. Since transmit RF power is much higher than receive power (typically eight orders of magnitude), signal detection is impossible during the ring-down time. In addition, the RF coil's recovery time also affects the rise time of the pulse [13]. For RF pulses of insufficient duration, long ring down times and slow rise times may not allow for the desired shape of the waveform, thereby affecting the applied flip angle. For example, in a spin echo sequence this would mean that the 90˚ and 180˚ pulses could not be applied appropriately for square RF pulses. The ring-down time can be shortened by either using a *Q*-switch [15–18], or through the application of a damping pulse which is 180˚ out-of-phase with main pulse [13, 18]. Similarly, the rise time can be shortened by applying pre-emphasis pulses with greater amplitude [13, 18].

Since the frequency profile of a sinc pulse is rectangular, it is often used for slice-selective excitation [19]. However, unlike with a rectangular pulse, the amplitude of the sinc pulse is not constant. When bandwidth of the sinc pulse is wide, its amplitude changes quickly. As a result, the actual pulse applied to the RF coil is often distorted and delayed. Waveform distortion is usually insignificant for the sinc pulse with narrow bandwidth but there can still be a delay in the applied RF pulse. The input pulse needed to compensate the sinc pulse can be calculated using the Laplace-transform of the impulse response function [20]. In this work, only time domain analysis was adapted to calculate the input pulse.

We calculated the duration of pre-emphasis and post-damping pulses for a given amplitude using the measured *Q*-factor. We also demonstrate a method for calculating an input pulse consisting of a series of square pulses designed to compensate the sinc pulse. For the

calculation of input pulses for compensating both square and sinc pulses, $Q$-factor was measured from ring-down time [21]. The waveforms of the compensated RF pulses were measured under different conditions and compared to the original RF pulses. MR signals were acquired by applying compensated pulse using a low field magnet with a non-uniform $B_0$ field and compared with the signal acquired by uncompensated pulse.

## Theory

### A. Square pulse

An RF coil can be modeled as an RLC series circuit and energy considerations in RLC circuit lead to a second-order differential equation:

$$L\frac{di}{dt} + Ri(t) + \frac{1}{C}\int_0^t i(\tau)d\tau = V(t).$$ (1)

where $L$, $R$, $C$, $i(t)$, and $V(t)$ are inductance, resistance, capacitance, the time-varying current, and applied time-varying voltage, respectively. When a square pulse (Fig 1a) is applied to an RF coil at low frequency, the amplitude of the resultant transmitted RF pulse is shown in Fig 1 (b) demonstrating a slow rise time rather than a rapid onset box-car shape. The amplitude of the actual pulse during $\tau_1$ is given by the solution of Eq 1.

$$A = A_1\left(1 - e^{-\frac{\omega}{2Q}\tau_1}\right).$$ (2)

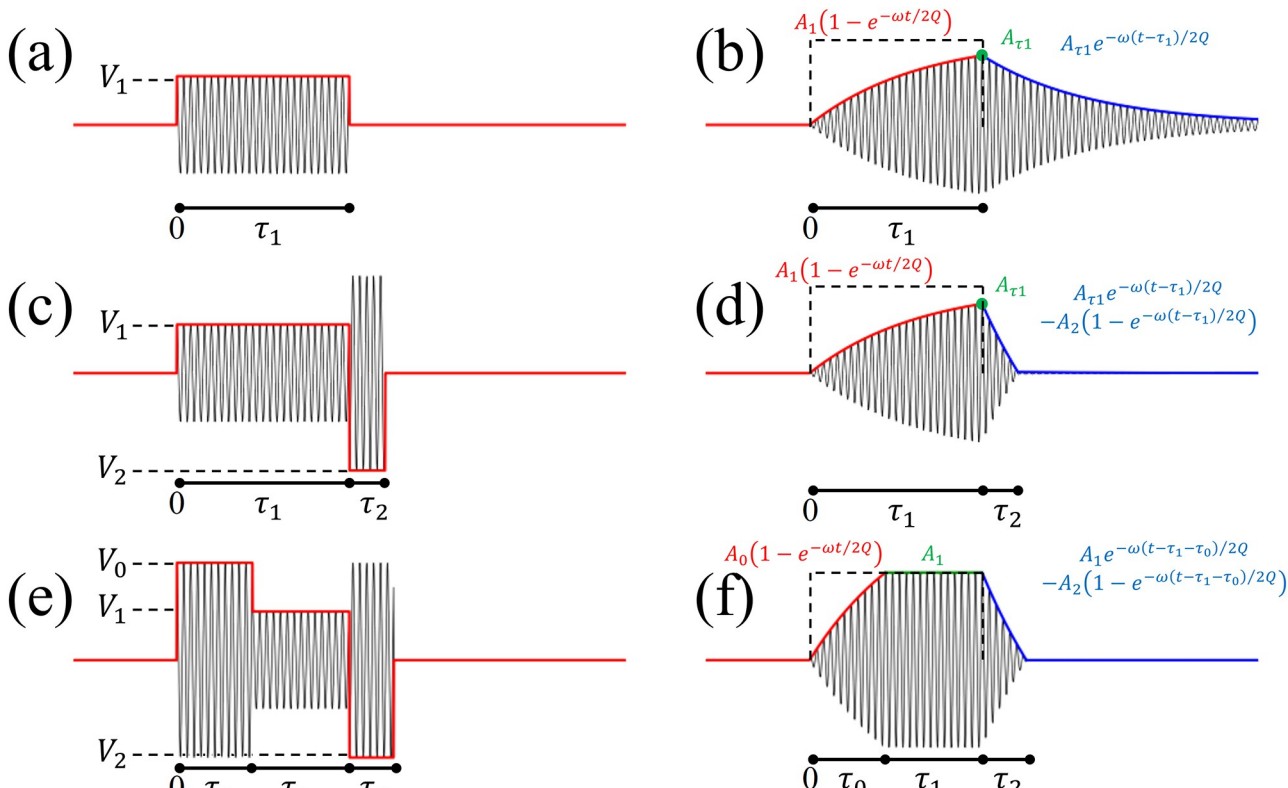

**Fig 1. Three different RF pulses applied in the power amplifier (left column) and the actual signal generated by the RF coil (right column).** (a) A square pulse and (b) its actual signal on the RF coil, (c) a square pulse with a post-pulse-damping lobe and (d) its actual signal on the RF coil, and (e) a square pulse with both pre-emphasis and a damping pulse and (f) its actual signal on the RF coil.

Where $A_1$, $\omega$, and Q are equilibrium amplitude of the RF signal, the resonance frequency, and the Q-factor of the RF coil, respectively. The $A_1$, and Q satisfy the formulas below.

$$A_1 = \frac{V}{Z}. \tag{3}$$

$$Q = \frac{\omega L}{R}. \tag{4}$$

where $Z$ ($Z = R + j(\omega L - 1/\omega C)$) is the total impedance of the RF coil. After switching off the input to the RF coil, there is also a long ring-down time before the signal dissipates. The amplitude of the RF signal ($A$) after switching off the pulse is given as the solution of Eq 1 with the condition of $V(t) = 0$.

$$A = A_{\tau 1} e^{-\frac{\omega}{2Q}(t-\tau_1)}, \tag{5}$$

Where $A_{\tau 1}$ is amplitude of the RF signal when the pulse is switched off at the time $\tau_1$. The RF ring-down time is reduced by applying additional square pulses that are 180° out-of-phase as shown in Fig 1c and 1d [13, 19]. If $\tau_1$ is long enough, $A_{\tau 1}$ reaches equilibrium ($A_1$) and the equilibrium amplitude is proportional to the applied voltage ($V_1$). For the given voltages for square pulse ($V_1$) and damping pulse ($V_2$), time ($\tau_2$) to zero amplitude can be calculated by the below equation.

$$V_1 - V_2 e^{-\frac{\omega}{2Q}(t-\tau_1)} = 0. \tag{6}$$

where $t = 0$. Thus, RF current is eliminated at time $\tau_1 + \tau_2$ 13].

$$\tau_2 = \frac{2Q}{\omega} \ln\left(1 + \frac{V_1}{V_2}\right). \tag{7}$$

However, with relatively short $\tau_1$, $A_{\tau 1}$ does not reach equilibrium and the time required to reach zero is rewritten as

$$\tau_2 = \frac{2Q}{\omega} \ln\left(1 + \frac{V_1}{V_2}\left(1 - e^{-\frac{\omega}{2Q}\tau_1}\right)\right). \tag{8}$$

Likewise, the transition time is shortened by applying pre-emphasis pulse with greater amplitude ($V_0$ s $V_1$) as shown in Fig 1e and 1f. $\tau_0$ is given as a function of $V_0$ and $V_1$.

$$\tau_0 = -\frac{2Q}{\omega} \ln\left(1 - \frac{V_1}{V_0}\right). \tag{9}$$

## B. Sinc pulse

The sinc pulse differs from the square pulse above in that the amplitude of the sinc pulse is not constant. But it too, can be compensated using multiple square pulses. The number of samples ($N_S$) in the signal can be defined as $N_S = 4iN_0 + 1$, where $N_0$ indicates the number of zero crossings on each side of the center before truncation, and $i$ is integer. Then, sampled points include all peaks and zero crossing points. For example, for the sinc with $N_0 = 3$ and $i = 1$, $N_S$ is equal to 13 and the peak and zero amplitude points are included in sampled points as shown in Fig 2.

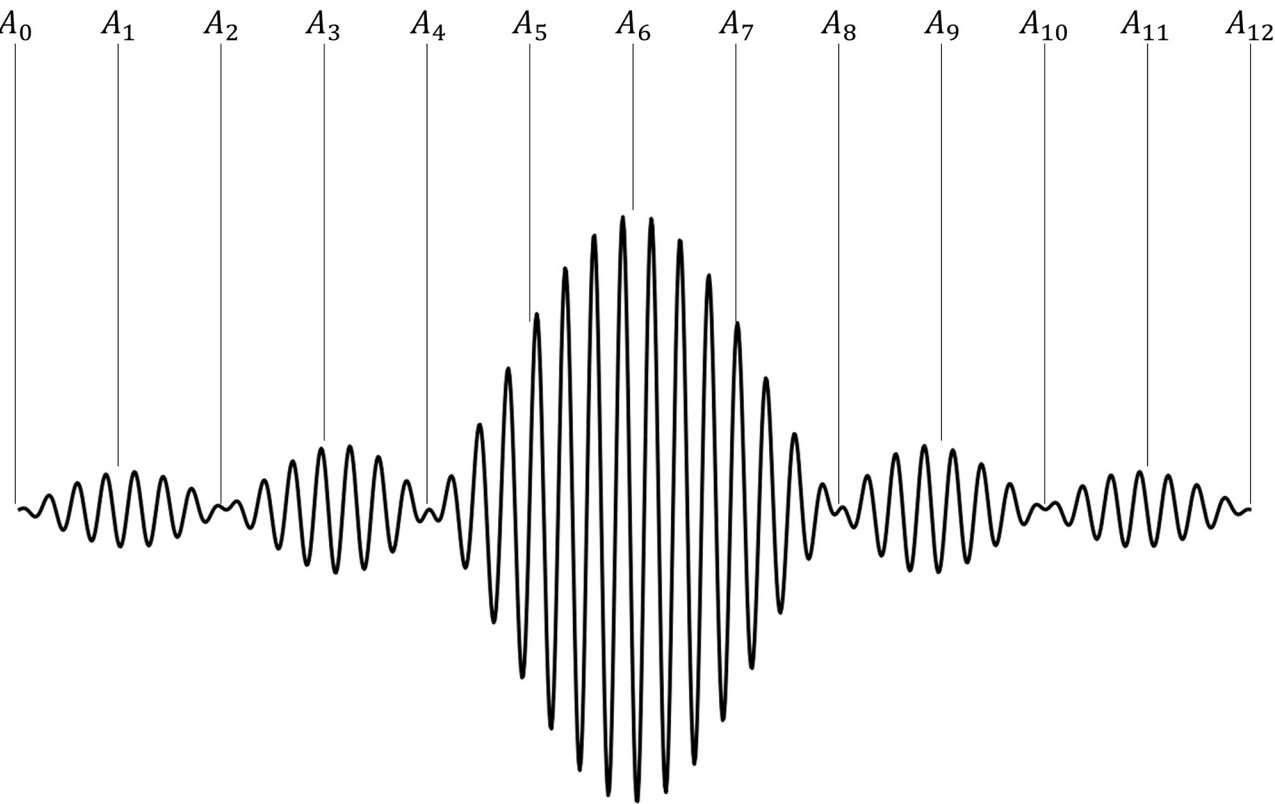

$$A_0 \quad A_1 \quad A_2 \quad A_3 \quad A_4 \quad A_5 \quad A_6 \quad A_7 \quad A_8 \quad A_9 \quad A_{10} \quad A_{11} \quad A_{12}$$

**Fig 2. Example of the sinc pulse ($N_0 = 3$, $i = 1$, $N_S = 13$).**

When a square pulse is applied with peak voltage of $V_1$ which satisfies the equation below, the amplitude of the actual signal is $A_1$ at $t_1$ ($t_n = n\Delta t$).

$$V_1 = \frac{A_1}{1 - e^{-\frac{\omega}{2Q}\Delta t}} \tag{10}$$

If a square RF pulse is not applied during the time interval between $t_1$ and $t_2$, the amplitude of the actual signal decays and can be calculated using Eq 5 at $t_2$. In order to make the amplitude to $A_2$ at $t_2$, a square pulse with peak voltage of $V_2$ should be applied to the RF coil. $V_2$ is defined as:

$$V_2 = \frac{A_2 - V_1\left(1 - e^{-\frac{\omega}{2Q}\Delta t}\right)e^{-\frac{\omega}{2Q}\Delta t}}{1 - e^{-\frac{\omega}{2Q}\Delta t}} \tag{11}$$

Similarly, the peak voltage ($Vn$) of the square pulse needed to compensate for the nth sampled signal is given by the following general form

$$V_n = \frac{A_n - \sum_{k=0}^{n-1} V_k\left(1 - e^{-\frac{\omega}{2Q}\Delta t}\right)e^{-\frac{\omega}{2Q}\Delta t(n-k)}}{1 - e^{-\frac{\omega}{2Q}\Delta t}} \tag{12}$$

As shown in Fig 3, smoother sinc pulses can be generated by increasing the number of samples, $N_S$.

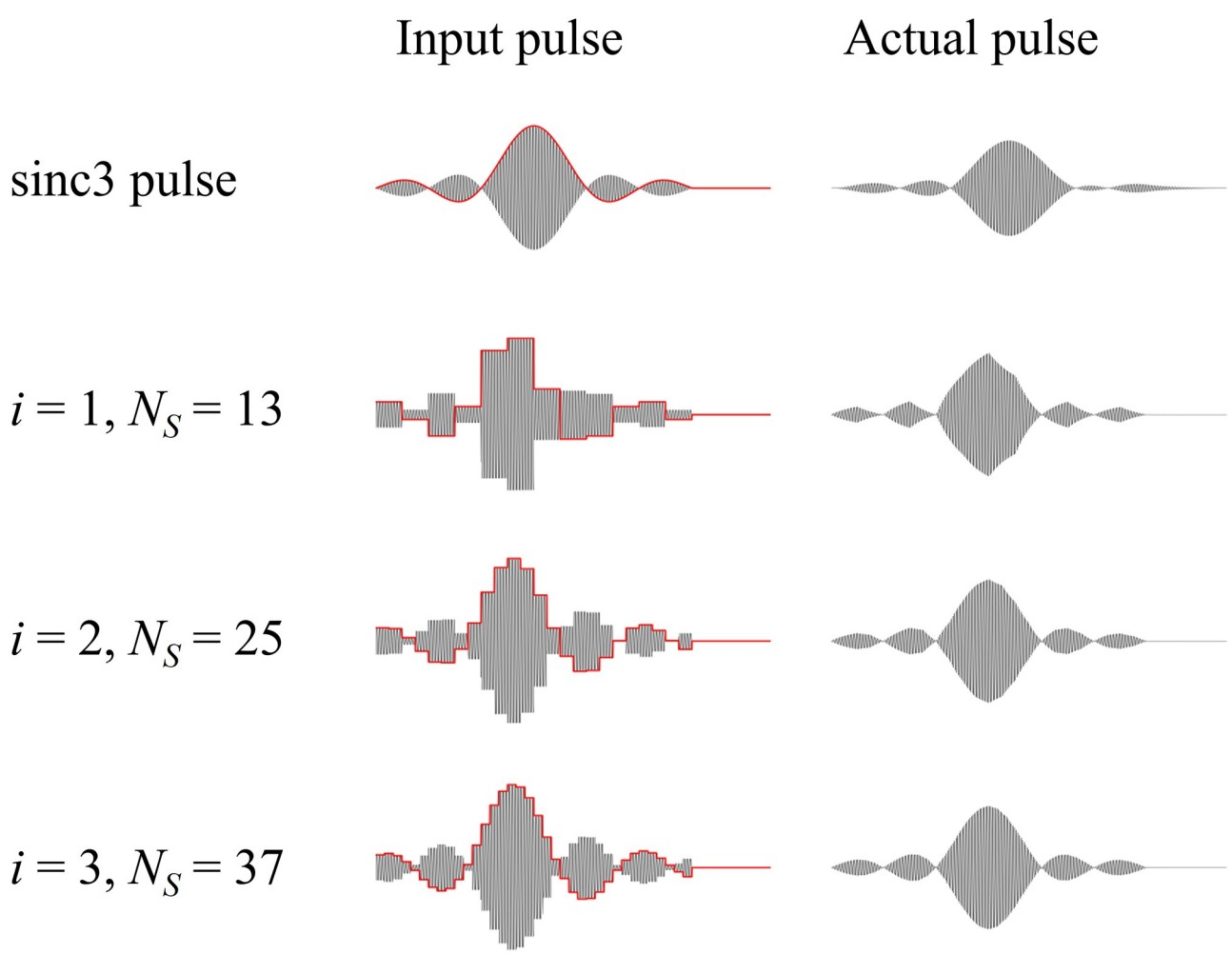

**Fig 3. Input RF pulses (left column) and the signal generated by the RF coil (right column) for sinc pulse (top row), calculated pulses with $N_S = 13$ (second row), $N_S = 25$ (third row), $N_S = 37$ (bottom row).**

## C. Q-factor measurement

As shown in the equations above, the $Q$-factor of the RF coil is needed to design compensation RF pulses. $Q$-factor is alternatively defined as the ratio of a resonance frequency to its bandwidth and the 3 dB bandwidth is defined as the frequency at which the power level of the signal decreases by half (3 dB) from its maximum value. $Q$-factor of the RF coil can be measured either using two decoupled sniffer loops (Fig 4a and 4b) [22] or calculated from the return loss of the RF coil (Fig 4c and 4d) [23]. When it is measured using two decoupled sniffer loops, both low and high cutoff frequency are measured at 3 dB down point from the peak. On the other hand, for the return loss method, those are measured at -3 dB points. This is because the half power of the signal is reflected at the –3 dB points while no signal is reflected at the peak with perfect matching condition. The $Q$-factor of the RF coil is given as

$$Q = \frac{f_r}{f_H - f_L} \tag{13}$$

where $f_r$, $f_L$, and $f_H$ are resonance frequency, lower cut-off frequency, and upper cut-off frequency, respectively.

## Experimental set up

## Outputs on VNA/Oscilloscope

**Fig 4. Three different methods for measuring Q-factor.** (a) Diagram using two decoupled sniffer loops and (b) example insertion loss between two sniffer loops. (c) Diagram to measure return loss of the RF coil and (d) example return loss of the RF coil. (e) Diagram to measure ring-down time and (f) example waveform of the applied RF pulse.

The distance between the coil and the sniffer loops affects the measurement. Similarly, the matching condition of the RF coil can affect the Q-factor when measured from the return loss of the coil. However, Q-factor of the coil can be measured from the ring-down time (Fig 4e and 4f) [21]. As shown in Fig 8f, the amplitude of the RF signal decays after switching off the RF pulse, and the rate of this decay is given by Eq 5. By counting how many cycles it takes to halve the amplitude ($N$), the Q-factor can be calculated as [21]:

$$Q = \frac{\pi N}{\ln(2)} \cong 4.53N. \tag{14}$$

## Methods

All experiments were carried out on an open, table-top, MRI scanner [2]. Fig 5a shows the schematic representation of the major system on scanner and the major interconnections.

This MRI system consists of two main components: a two-element electromagnet (Fig 5c), and an RF system. Since the electromagnet generates a non-uniform polarizing $B_0$ field, slice selective excitation (exciting nonuniform isofrequency bands) can be performed without using gradient coils. The maximum $B_0$ field was 300 mT at the bottom of the magnet, and 200 mT at a height of 10 cm. The RF solenoid coil is tuned at 1 MHz (23.4 mT) and generates an alternating $B_1$ magnetic field which is perpendicular to $B_0$ (Fig 5b). The RF coil also detects the NMR signal created by the sample. The RF pulse was generated by a spectrometer (Redstone,

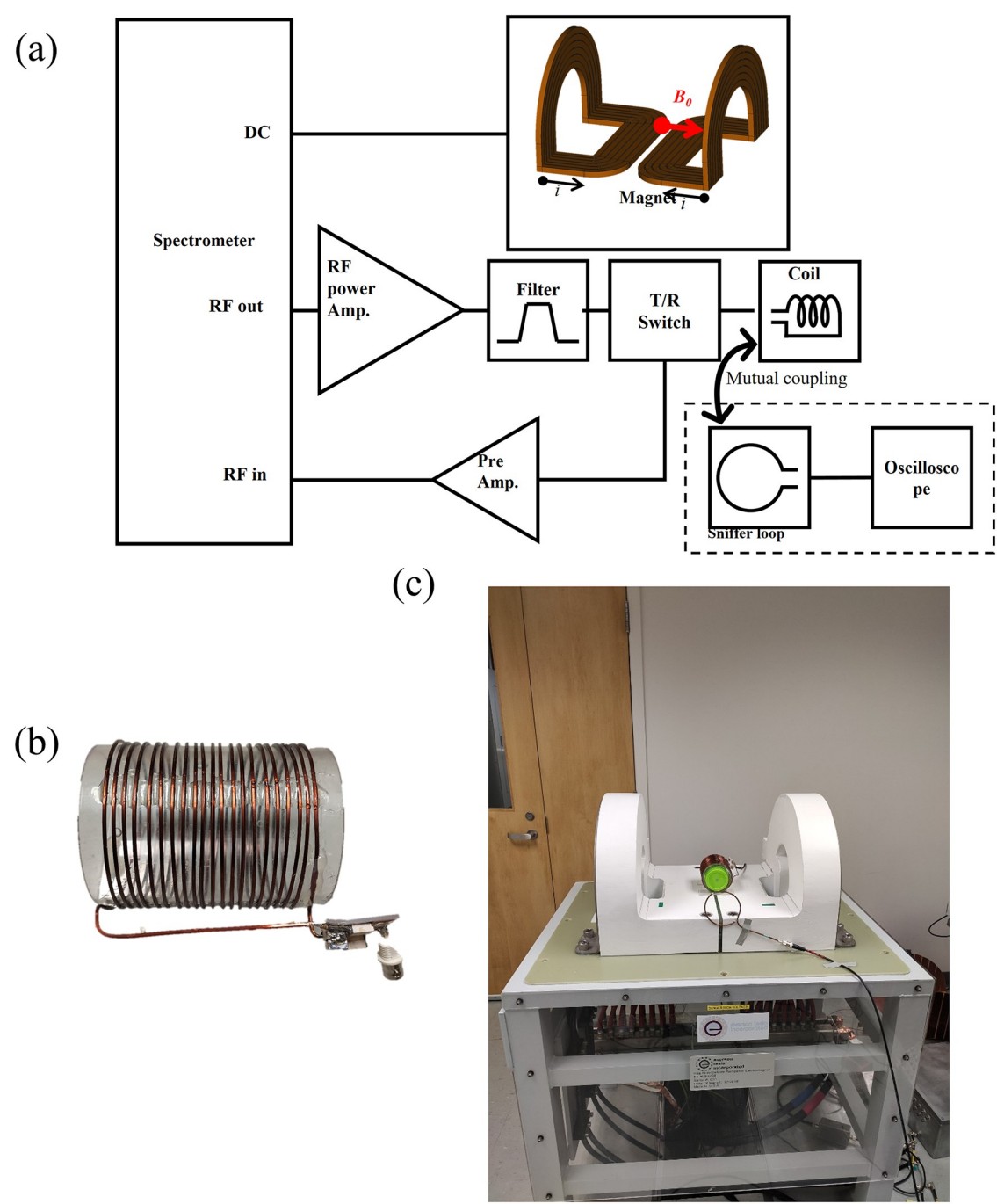

**Fig 5. Table-top MRI system.** (a) Schematic diagram of an open, table-top, MRI system with photos of (b) the RF solenoid coil and (c) the magnet.

Tecmag Inc., TX, USA), and amplified using an RF power amplifier (AR 0.5-15-1E-3-3C, CPC, NY, USA). A band pass filter built in-house was used to eliminate any harmonics in the applied RF pulse. A passive T/R switch with crossed-diode pair was also used.

A 40 μs square pulse with an amplitude of 80.2 mV was applied to the RF coil, and the resulting signal measured using a sniffer loop connected to an oscilloscope (Picoscope 4824,

Pico technology, UK). From the ring-down signal of the measured pulse, the $Q$-factor was calculated using Eq 14. Using the $Q$-factor measured from ring-down time, $\tau_0$, and $\tau_2$ of the compensation pulse were calculated. The compensated square pulses were then applied to the RF coil, and the resulting signal was also measured using an oscilloscope.

Sinc pulses with different bandwidth (30 kHz, 90 kHz, and 150 kHz) were also applied to the RF coil. The actual signal and frequency spectrums were measured using an oscilloscope. The amplitude and duration for the compensated sinc pulse were calculated for different $N_S$ (13, 25, and 37) using Python (Python Software Foundation). Then, compensated sinc pulses were applied to the RF coil and the actual pulse frequency spectrums were measured and compared using a sniffer loop connected to an oscilloscope. For this purpose, relative error was calculated using Eq 15 by dividing the total sum of difference between magnitude and mean of the magnitude through the total points within the bandwidth.

$$Relative\ error = \frac{\sum_{within\ the\ bandwidth}(magnitude - mean(magnitude))}{Total\ points}. \tag{15}$$

### Phantom experiments

$^1$H echo trains of a uniform cylindrical phantom (110 mm length and 70 mm diameter, distilled water) were obtained using compensated square and sinc pulses. Fig 6 shows the diagram of applied sequence.

All the scan parameters used in this study are described in Table 1. For the square pulse experiment, 20 μs and 40 μs of pulse duration was applied for 90° and 180° pulses, respectively. For the scan with compensated sinc pulse, a pulse with double amplitude of 90° pulse was applied as the 180° pulse. The SNR of the NMR signal was also compared with and without using of RF pulse distortion compensation. For the SNR calculations, the signal mean (maximum intensity) of the first 20 echoes was divided by the noise standard deviation of the last 20 echoes (the NMR signal at the last 20 echoes had been decayed).

### Results

Fig 7 shows the measured square input pulse without (a) and with (b) a compensated input.

$Q$-factor was calculated using Fig 7a and found to be 49.83. For the square pulse, the amplitudes and durations of the compensation pulse were found to be 120.3 mV, 40.1 mV, 120.3 mV, 4.68 μs, 40 μs. and 3.32 μs for $V_0$, $V_1$, $V_2$, $\tau_0$, $\tau_1$, and $\tau_2$, respectively.

Fig 8 shows the calculated amplitude for compensating the sinc pulses for the different bandwidths.

Since the amplitude changes rapidly for the wide band pulse, higher amplitudes are required for the compensation. Fig 9 shows the measured actual pulses when sinc and compensated sinc pulses were applied.

The long response time of the RF coil also affects the shape of the sinc pulse, especially for broad band pulses. The compensation helps with both reduction of the amplitude after the input pulse end and frequency profile. Table 2 shows the relative errors of the signal magnitude of the frequency spectrum.

Fig 10 shows the $^1$H NMR signal from a water phantom. The $x$-axis shows time in microseconds and y axis shows the normalized signal intensity.

The uncompensated and the compensated square pulses shown in Fig 8 was used as a 90° excitation pulse followed by a series of 180° refocusing RF pulses (the same pulse doubled in duration) in order to acquire the echo train. Fig 10 (third and bottom rows) shows the $^1$H

Pre-polarization

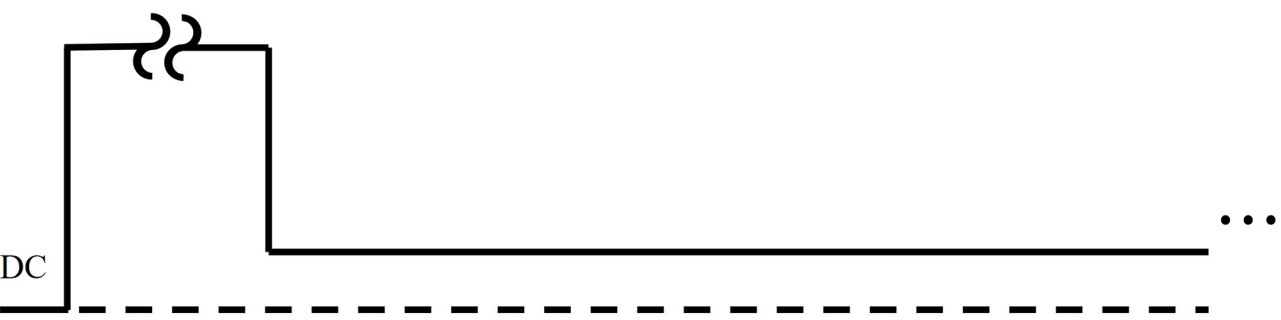

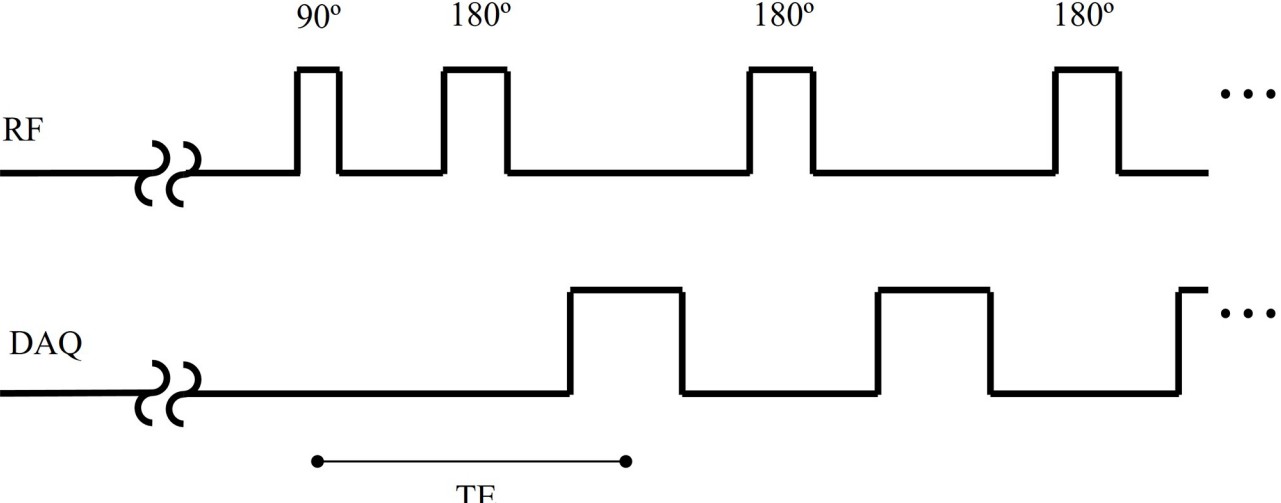

**Fig 6. A spin-echo sequence with pre-polarization.**

NMR signal of water phantom acquired using uncompensated and compensated sinc pulses (the same pulse doubled in amplitude) which was calculated with $N_S = 37$. Table 3 summarizes the SNR values of the NMR signals.

## Discussion

When the Q-factor of the RF coil is high, the bandwidth of the RF coil can be narrower than the bandwidth of the applied RF pulse. For this reason, especially in low magnetic field MRI

**Table 1. NMR signal acquisition parameters used in this study.**

|  | Pre-polarization time | Dwell time | Sampling time | TR | TE | number of echoes | Averages |
|---|---|---|---|---|---|---|---|
| Square pulse | 5 s | 10 μs | 320 μs | ~ 6 s | 1 ms | 1,000 | 32 |
| Sinc pulse |  |  |  | ~ 6.5 s | 1.5 ms | 1,000 | 32 |

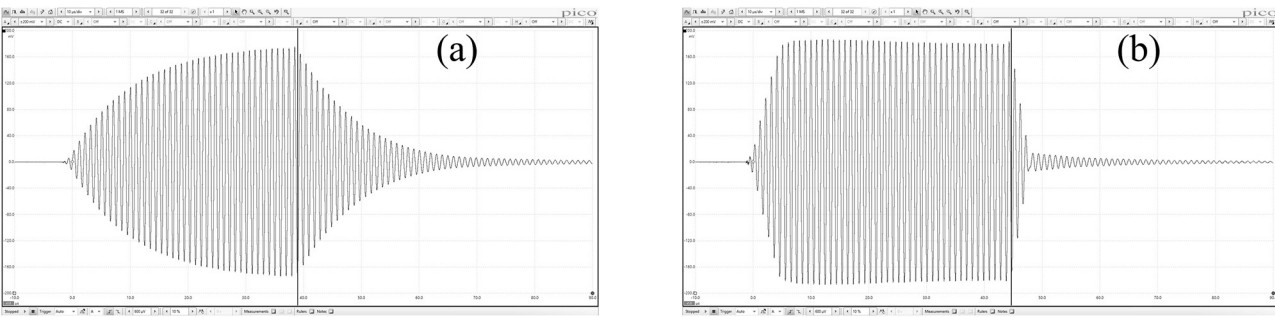

**Fig 7. Measured actual pulses of (a) square pulse input and (b) compensated square pulse input.** Vertical solid lines indicate the input pulse ends.

(low resonance frequency), short RF pulses can be easily distorted due to the narrow bandwidth of the RF coil. This study demonstrates that compensation pulses (including the required duration and amplitude of such a pulse) can be calculated utilizing the measured $Q$-factor, and these compensation pulses lead to RF shapes closer to the desired output. The SNR of the NMR signal acquired by compensated RF pulses were compared with those obtained by using uncompensated RF pulses.

For the square RF pulse, three square pulses were applied rather than applying a single square RF pulse [13]. While the amplitude of a square pulse only changes once at the

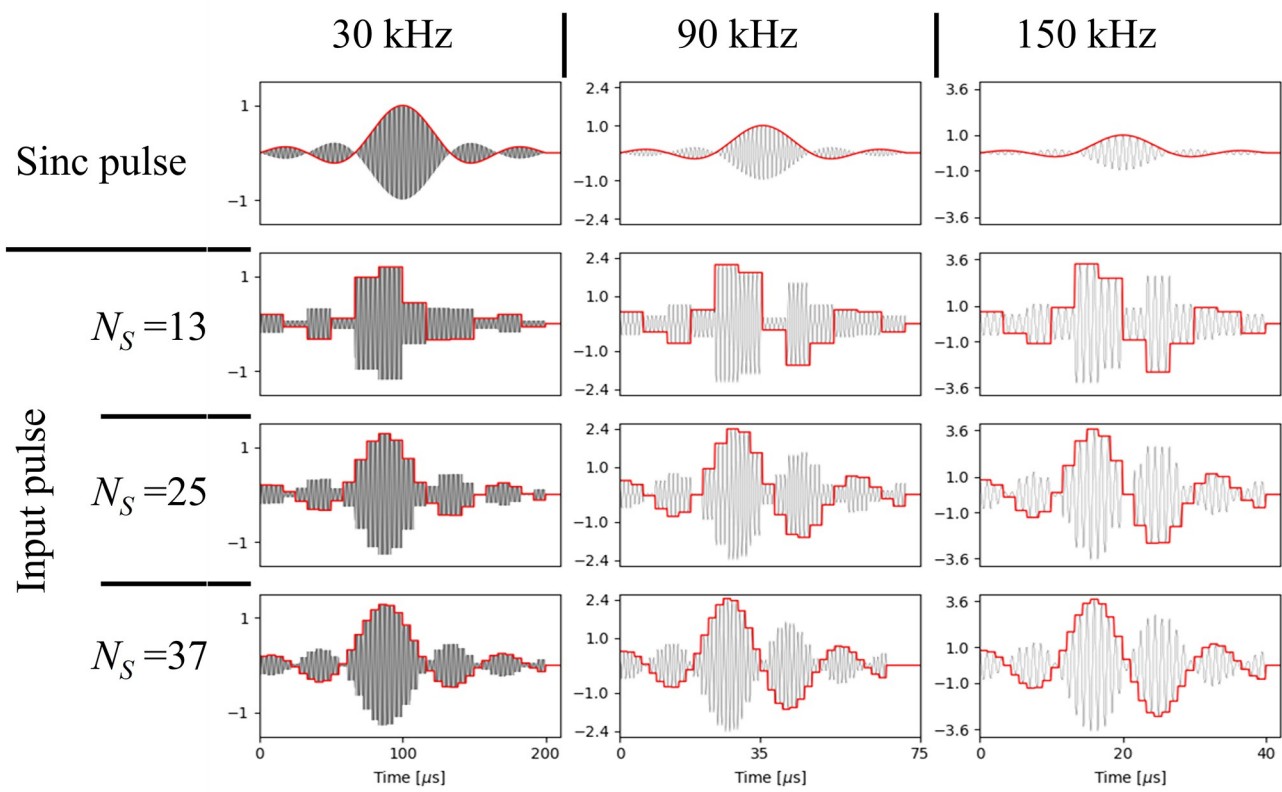

**Fig 8. Sinc pulses (top row) and calculated input pulses for compensation calculated with different $N_S$ (13 (second row), 25 (third row), and 37 (bottom row)) for different bandwidth (30 kHz (left column), 90 kHz (middle column), and 150 kHz (right column)).**

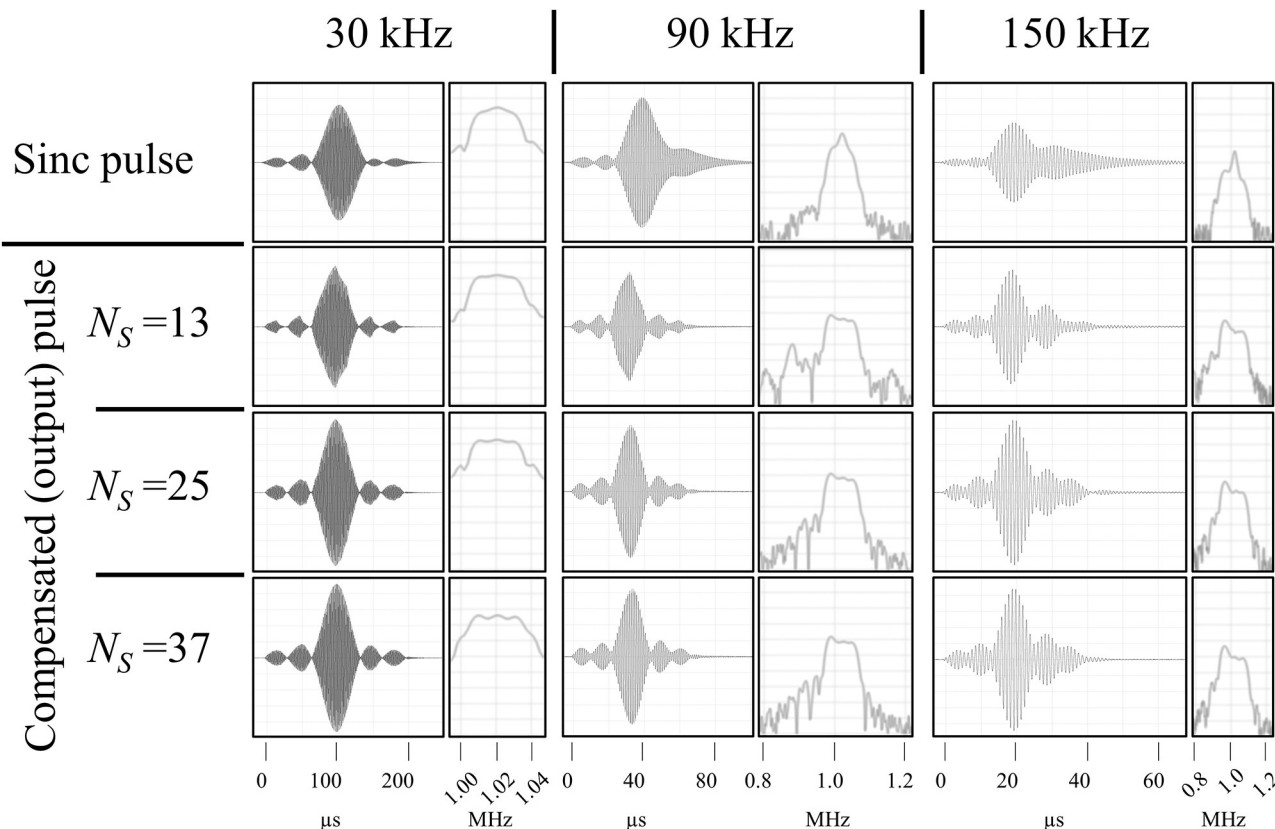

**Fig 9. Measured actual pulses (output pulses, vertical dashed lines indicate the input pulse ends).** And frequency spectrums of the sinc pulses (top row) and compensated sinc pulses calculated with $N_S$ values of 13 (second row), 25 (third row), and 37 (bottom row) and bandwidths of 30 kHz (left column), 90 kHz (middle column), and 150 kHz (right column). The y-axis shows normalized values (from -1 to 1 for waveforms, from 0 to 1 for frequency spectrums).

beginning and once at the end of the pulse, the signal amplitude of a sinc pulse changes throughout the duration of the pulse. These amplitude changes can cause time delays and waveform distortions as they propagate through the coil. It was demonstrated that the distortion of a sinc pulse can be compensated by using series of square pulses, and that a smoother waveform can be acquired by using more square pulses with short durations. Furthermore, the frequency profiles of the compensated sinc pulses were closer to a square than that of the original sinc pulses especially for the wide bandwidth. The time delay of the pulse was also compensated using this method, which is helpful if a hardware $Q$-switch is used. If a $Q$-switch is used without compensation, the end of applied pulse can be truncated due to the time delay. Although only a square pulse and sinc pulse, the latter with N0 = 1 to 3, were considered here,

**Table 2. Relative errors of the signal magnitude of the frequency spectrum.**

| Bandwidth | | Relative error | | |
|---|---|---|---|---|
| | | **30 kHz** | **90 kHz** | **150 kHz** |
| Sinc pulse | | 0.33 | 0.51 | 0.62 |
| Compensated pulse | $N_S = 13$ | 0.25 | 0.23 | 0.24 |
| | $N_S = 25$ | 0.27 | 0.23 | 0.23 |
| | $N_S = 37$ | 0.26 | 0.23 | 0.24 |

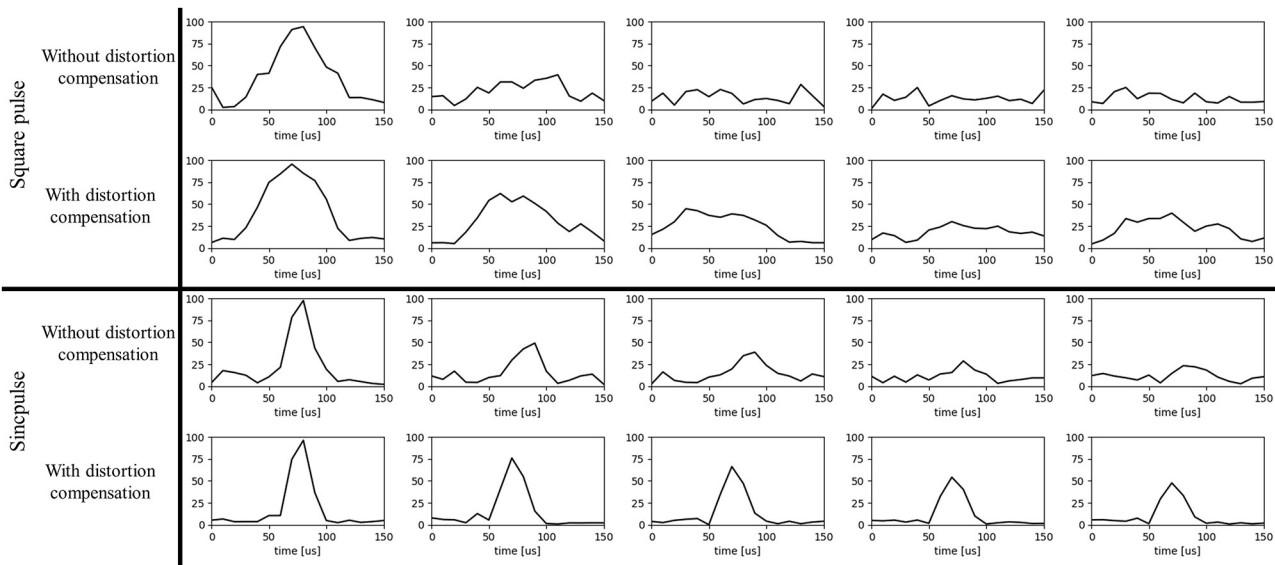

**Fig 10.** [1]H 25[th], 50[th], 75[th], 100[th], and 125[th] echoes from a train of 1000 echoes, of water obtained using square pulse (top row), compensated square pulse (second row), sinc pulse with 90 kHz bandwidth (third row), and compensated sinc pulse, 90 kHz bandwidth, n = 37 (bottom row).

the suggested method can be used for any shape of transmit pulses such as sinc1, sinc5, and Gaussian pulses. As demonstrated here both the square and the sinc pulse shapes can be improved by calculating compensatory input pulses using $Q$-factor. Thus, it is important to accurately measure the $Q$-factor. The $Q$-factor of the RF coil is usually measured either using two decoupled sniffer loops [22] or from a measurement of the return loss [23] of the RF coil. However, the $Q$-factor measured from the ring-down time is different from the $Q$-factor measured using vector network analyzer since other factors such as the distance between the RF coil and sniffer loops and the matching condition of the RF coil can influence the results. Moreover, RF recovery time is also affected by the RF chain (filter, T/R switch, and co-axial cable) in addition to the RF coil itself. This $Q$-factor measurement worked well for generating RF compensation pulses here. For the higher precision of $Q$-factor, $N$ in Eq 14 can be estimated using decimal rather than using only integer values.

Since the $Q$ value is directly related to the bandwidth, the signal distortion had a greater effect when a wide bandwidth sin pulse was applied. This was confirmed by the waveform, especially the wider the bandwidth, the longer the ring down time. It was confirmed that this long ring-down signal was effectively reduced through the proposed method. Through the frequency profile, it is confirmed that the wide bandwidth signal is more distorted, and the proposed method has a correction effect. It was also confirmed through the calculated relative error.

By applying compensated pulses (to both square and sinc pulses), multi-echo spin echo NMR signal were acquired on inhomogeneous magnetic field. A CPMG spin echo sequence

**Table 3. SNR values measured by using uncompensated and compensated pulses (square and sinc pulses).**

|  | SNR | |
| --- | --- | --- |
|  | **Without distortion compensation** | **With distortion compensation** |
| Square pulse | 9.9 | 16.2 |
| Sinc pulse | 14.0 | 27.2 |

was used to acquire 1000 spin-echoes at low magnetic field (1 MHz) after the spins had been pre-polarized at approximately 300 mT (12.77 MHz) [3–5] in a novel field cycling open MR system. As shown in Fig 9, the output pulse can excite more spins within the slice to the desired flip angle. As a result, the SNR of the echo signals obtained using compensated pulses were 61.1% and 51.5% higher relative to that obtained using uncompensated pulses for square pulse and sinc pulse, respectively, indicating substantial improvement with this compensation strategy.

In conclusion, we have introduced an approach to low field imaging for measuring the $Q$-factor of an RF coil, in order to derive a set of compensation pulses to reduce distortions in the shape of the excitation profile. For the accurate calculation of the compensating pulses, the $Q$-factor was measured from the ring-down time of the coil. It was demonstrated that both square and sinc pulse shapes could be improved by applying compensating RF pulses at the RF input. Echo trains 1000 echoes long were acquired using these pulses at low readout field $B_0$. Improvement in the echo SNR was demonstrated by using RF pulse compensation for both square and sinc pulses.

## Supporting information

**S1 Data.**
(PY)

## Author Contributions

**Conceptualization:** Yonghyun Ha, R. Todd Constable.

**Funding acquisition:** Gigi Galiana, R. Todd Constable.

**Methodology:** Yonghyun Ha, Kartiga Selvaganesan, Baosong Wu, Kasey Hancock, Charles Rogers, III, Sajad Hosseinnezhadian.

**Project administration:** Gigi Galiana, R. Todd Constable.

**Supervision:** Gigi Galiana, R. Todd Constable.

**Validation:** Kartiga Selvaganesan.

**Writing – original draft:** Yonghyun Ha.

**Writing – review & editing:** Yonghyun Ha, Kartiga Selvaganesan, Baosong Wu, Kasey Hancock, Charles Rogers, III, Sajad Hosseinnezhadian, Gigi Galiana, R. Todd Constable.

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
