## [Decision Letter · Decision Letter 0]

24 May 2022

PONE-D-22-10094Practical method for RF pulse distortion compensation using multiple square pulses for low-field MRIPLOS ONE

Dear Dr. Constable,

Thank you for submitting your manuscript to PLOS ONE. After careful consideration, we feel that it has merit but does not fully meet PLOS ONE’s publication criteria as it currently stands. Therefore, we invite you to submit a revised version of the manuscript that addresses the points raised during the review process.

We look forward to receiving your revised manuscript.

Kind regards,

Yefeng Yao

Academic Editor

PLOS ONE

Journal Requirements:

Reviewers' comments:

Reviewer's Responses to Questions

**Comments to the Author**

1. Is the manuscript technically sound, and do the data support the conclusions?

Reviewer #1: Yes

Reviewer #2: Yes

2. Has the statistical analysis been performed appropriately and rigorously? 

Reviewer #1: I Don't Know

Reviewer #2: Yes

3. Have the authors made all data underlying the findings in their manuscript fully available?

Reviewer #1: Yes

Reviewer #2: Yes

4. Is the manuscript presented in an intelligible fashion and written in standard English?

Reviewer #1: Yes

Reviewer #2: Yes

5. Review Comments to the Author

Reviewer #1: This manuscript proposed a method for RF pulse distortion compensation using multiple square pulses for low-field MRI. The authors demonstrated that compensation pulses could be calculated utilizing the measured Q-factor, and these compensation pulses leaded to RF shapes closer to the desired output. The results are interesting and worthy of publication. The manuscript can be improved by addressing the following concerns.

(1) The quality of figures in the manuscript should be improved. The resolutions of the figures are too low.

(2) I suggest that the authors provide more information for the phantom and more scanning parameters for spin-echo sequence (receiver bandwidth, sampling points, etc.).

(3) The method of calculating SNR of signal should be provided.

(4) The authors should explain why the SNRs were improved after applying the RF with distortion compensation.

(5) I suggest that the authors discuss the advantages of the proposed method for human body low-field MRI. In human body low-field MRI, two coils are used for RF transmission and signal receive, respectively. The duration of RF pulse is usually several milliseconds. Is it necessary to use the proposed method?

(6) Minor concerns

Page#15, Line#15: 180s ->180° refocused RF pulses

Reviewer #2: This work introduces a RF square pulse compensated method by appending two square pulses before and after the RF pulse, with durations of these RF square pulses calculated using the Q-factor. It was also successfully demonstrated that the sinc pulse can be compensated using a series of square pulses. The more number of square pulses were used, the smoother sinc pulse was applied to the RF coil. Echo trains were also acquired in an inhomogeneous B0 field using the compensated RF pulses. In order to enhance the SNR of the echo trains, a pre-polarization pulse was added to the CPMG spin echo sequence. The SNRs of the echo signal acquired using compensated pulses were compared with those of signal obtained with uncompensated pulses and showed significant improvements of 61.1% and 51.5% for the square and sinc shaped pulses respectively.

Some questions are as follows:

Page 15, line 4, table 2, why the sinc pulse deviation for larger radiofrequency is bigger comparing lower frequency condition, which seems contradictory with equation 2.

In fig.1, how to control the phase of the pre-emphasis pulse(figure 1c)&e)) to keep the phase of the pulse constant?

In fig.8, why the sinc pulse in Ns =25&37 are asymmetric after the compensation? while this also appears in fig.9.

In fig.10, the axis in the figure are vague, please make it more readable.

This work is interesting and applicable in low-field NMR/MRI.

6. PLOS authors have the option to publish the peer review history of their article (what does this mean?). If published, this will include your full peer review and any attached files.

Reviewer #1: **Yes: **Jianqi Li

Reviewer #2: **Yes: **Xiaodong YANG

---

## [Author Response · Author response to Decision Letter 0]

27 Jun 2022

Reviewer #1: This manuscript proposed a method for RF pulse distortion compensation using multiple square pulses for low-field MRI. The authors demonstrated that compensation pulses could be calculated utilizing the measured Q-factor, and these compensation pulses leaded to RF shapes closer to the desired output. The results are interesting and worthy of publication. The manuscript can be improved by addressing the following concerns.

(1) The quality of figures in the manuscript should be improved. The resolutions of the figures are too low.

We have regenerated figures (Fig. 2, 4, and 10). The resolution of the figures in the pdf file looks low. However, high resolution images are available by clicking the links (downloading the figures) in the manuscript document.

(2) I suggest that the authors provide more information for the phantom and more scanning parameters for spin-echo sequence (receiver bandwidth, sampling points, etc.).

 We have added details about phantom as:

“H echo trains of a uniform cylinder phantom (110 mm length and 70 mm diameter, distilled water) were obtained using compensated square and sinc pulses.”

We have also added dwell time (10 us) and sampling time (320 us) to table 1, so receiver bandwidth and sampling points are provided.

(3) The method of calculating SNR of signal should be provided.

 We have added details about SNR calculation as:

“For the SNR calculations, the signal mean (maximum intensity) of the first 20 echoes was divided by the noise standard deviation of the last 20 echoes (the NMR signal at the last 20 echoes had decayed to essentially zero).”

(4) The authors should explain why the SNRs were improved after applying the RF with distortion compensation.

When a distorted signal is applied, some spins within the slice are excited with lower flip angle than expected. This can be corrected through the proposed compensation method, resulting in increased SNR. We have added details in discussions as:

As shown in Fig. 8, the calculated input pulses for distortion compensation require higher power. However, as shown in Fig. 9, the output pulse can excite more spins within the slice to the desired flip angle. As a result, the SNR of the echo signals obtained using compensated pulses were 61.1 % and 51.5 % higher relative to that obtained using uncompensated pulses for square pulse and sinc pulse, respectively, indicating substantial improvement with this compensation strategy.

(5) I suggest that the authors discuss the advantages of the proposed method for human body low-field MRI. In human body low-field MRI, two coils are used for RF transmission and signal receive, respectively. The duration of RF pulse is usually several milliseconds. Is it necessary to use the proposed method?

RF distortion may not be a problem in human body low-field MRI with several milliseconds duration of RF pulse. However, it depends on not only the field strength (carrier frequency) and the pulse duration (BW of the pulse), but also bandwidth (Q) of the RF coil as described in Eq. 2 and Eq. 5. We have modified a sentence in the introduction section as:

When the Q-factor of the RF coil is high, the bandwidth of the RF coil can be narrower than the bandwidth of the applied RF pulse. For this reason, especially in low magnetic field MRI (low resonance frequency), short RF pulses can be easily distorted due to the narrow bandwidth of the RF coil (long RF coil recovery times).

(6) Minor concerns

Page#15, Line#15: 180s ->180° refocused RF pulses

 Thank you for your careful proofreading. We have made the suggested change.

Reviewer #2: This work introduces a RF square pulse compensated method by appending two square pulses before and after the RF pulse, with durations of these RF square pulses calculated using the Q-factor. It was also successfully demonstrated that the sinc pulse can be compensated using a series of square pulses. The more number of square pulses were used, the smoother sinc pulse was applied to the RF coil. Echo trains were also acquired in an inhomogeneous B0 field using the compensated RF pulses. In order to enhance the SNR of the echo trains, a pre-polarization pulse was added to the CPMG spin echo sequence. The SNRs of the echo signal acquired using compensated pulses were compared with those of signal obtained with uncompensated pulses and showed significant improvements of 61.1% and 51.5% for the square and sinc shaped pulses respectively.

Some questions are as follows:

Page 15, line 4, table 2, why the sinc pulse deviation for larger radiofrequency is bigger comparing lower frequency condition, which seems contradictory with equation 2.

Equation 2 describes the rise time of the applied pulse. This shows that the signal takes time to increase in magnitude. The wide bandwidth of a sine pulse means that the amplitude of the signal changes rapidly. The wider the bandwidth, the greater the signal distortion, because the change in signal amplitude of a wide bandwidth sinc pulse is shorter than the time required for the change in signal amplitude. 

To avoid confusion, we added that frequency represents bandwidth in Table 2.

In fig.1, how to control the phase of the pre-emphasis pulse(figure 1c)&e)) to keep the phase of the pulse constant?

Ring-down is not affecting the carrier frequency but the envelope of the pulse. However, during the τ2, a square pulse with 180º out-of-phase is applied to cancel out the ring-down signal. 

In fig.8, why the sinc pulse in Ns =25&37 are asymmetric after the compensation? while this also appears in fig.9.

Fig. 8 shows the calculated input pulse for compensation. By applying these calculated asymmetric input pulses to the RF coil, compensated sinc pulses were generated. When the input pulse is a sinc (symmetric) shape, the applied pulse can be asymmetrically distorted as shown in Fig. 9 first row.

To avoid confusion, we modified the figure caption of Fig. 9 as:

Measured actual pulse (vertical dashed lines indicate the input pulse ends.) → Measured actual pulse (output pulses, vertical dashed lines indicate the input pulse ends.)

In fig.10, the axis in the figure are vague, please make it more readable.

Thank you for your careful proofreading. 

We have adjusted the figure size.

This work is interesting and applicable in low-field NMR/MRI.

---

## [Decision Letter · Decision Letter 1]

9 Aug 2022

Practical method for RF pulse distortion compensation using multiple square pulses for low-field MRI

PONE-D-22-10094R1

Dear Dr. Constable,

We’re pleased to inform you that your manuscript has been judged scientifically suitable for publication and will be formally accepted for publication once it meets all outstanding technical requirements.

Kind regards,

Yefeng Yao

Academic Editor

PLOS ONE

Additional Editor Comments (optional):

Reviewers' comments:

Reviewer's Responses to Questions

**Comments to the Author**

1. If the authors have adequately addressed your comments raised in a previous round of review and you feel that this manuscript is now acceptable for publication, you may indicate that here to bypass the “Comments to the Author” section, enter your conflict of interest statement in the “Confidential to Editor” section, and submit your "Accept" recommendation.

Reviewer #1: All comments have been addressed

Reviewer #2: All comments have been addressed

2. Is the manuscript technically sound, and do the data support the conclusions?

Reviewer #1: Yes

Reviewer #2: Yes

3. Has the statistical analysis been performed appropriately and rigorously? 

Reviewer #1: Yes

Reviewer #2: Yes

4. Have the authors made all data underlying the findings in their manuscript fully available?

Reviewer #1: Yes

Reviewer #2: Yes

5. Is the manuscript presented in an intelligible fashion and written in standard English?

Reviewer #1: Yes

Reviewer #2: Yes

6. Review Comments to the Author

Reviewer #1: (No Response)

Reviewer #2: The authors have well addressed my questions, and the current version is fine for me. The method and result in the paper is attractive for the readers, i believe.

7. PLOS authors have the option to publish the peer review history of their article (what does this mean?). If published, this will include your full peer review and any attached files.

Reviewer #1: **Yes: **Jianqi Li

Reviewer #2: No

---

## [Editor Report · Acceptance letter]

7 Sep 2022

PONE-D-22-10094R1 

Practical method for RF pulse distortion compensation using multiple square pulses for low-field MRI 

Dear Dr. Constable:

I'm pleased to inform you that your manuscript has been deemed suitable for publication in PLOS ONE. Congratulations! Your manuscript is now with our production department. 

Kind regards, 

on behalf of

Dr. Yefeng Yao 

Academic Editor

PLOS ONE